# Cost-effectiveness of percutaneous patent foramen ovale closure versus medical therapy for cryptogenic stroke prevention: A Chinese healthcare perspective

Chen Chen[1☯], Ting Xu[2☯], Zhujun Zeng[3☯], Jing Miao[1], Rong Hu[1], Yiming Gao[1], Chunlin Li[1], Meng Li[1], Xiaoli Yang [2]*

1 Department of Cardiac Surgery, Huashan Hospital, Fudan University, Shanghai, China, 2 Department of Infection Control, Huashan Hospital, Fudan University, Shanghai, China, 3 Department of Gastroenterology, Huashan Hospital, Fudan University, Shanghai, China

☯ These authors contributed equally to this work.
* xiaoliyang@fudan.edu.cn

## Abstract

### Background

Patent foramen ovale occurs in approximately 25% of cryptogenic stroke patients, with paradoxical embolization representing a key pathophysiological mechanism. While recent trials have established the clinical efficacy of percutaneous PFO closure in reducing recurrent stroke risk, comprehensive economic evaluation within the Chinese healthcare framework remains limited. This study assessed the cost-effectiveness of PFO closure compared with medical therapy alone in Chinese patients with cryptogenic stroke.

### Methods

A Markov cohort model was constructed with four health states (stable, post-minor recurrent stroke, post-moderate-to-severe recurrent stroke, death) using 3-month cycles over a 30-year horizon. The analysis adopted a Chinese healthcare payer perspective, incorporating transition probabilities derived primarily from the RESPECT trial extended follow-up data and Chinese-specific cost data from multi-center hospital surveys. Primary outcome was the incremental cost-effectiveness ratio (ICER) evaluated against China's willingness-to-pay threshold of $37,654 per quality-adjusted life year (QALY). Extensive sensitivity analyses examined parameter uncertainty and model robustness.

### Results

PFO closure demonstrated economic dominance over medical therapy, yielding 1.41 additional QALYs (13.42 vs 12.01) while reducing lifetime costs by $4,045 per patient

**Data availability statement:** All relevant data are within the manuscript and its Supporting Information files.

**Funding:** This work was supported by the Fudan-Fosun Research Fund (grant number FNF202329). The funder had no role in study design, data collection and analysis, decision to publish, or preparation of the manuscript.

**Competing interests:** The authors have declared that no competing interests exist.

($8,847 vs $12,892). The resulting negative ICER of -$2,868 per QALY indicated superior outcomes at lower cost. Probabilistic sensitivity analysis revealed 94.2% probability of cost-effectiveness at the Chinese threshold. One-way sensitivity analyses consistently showed negative ICERs across all parameter ranges, with time horizon and psychological comorbidity costs exerting the greatest influence on results.

## Conclusions

In the Chinese healthcare context, PFO closure represents a dominant economic strategy for cryptogenic stroke patients, providing both clinical benefits and cost savings. These findings support broader implementation of PFO closure programs and inform evidence-based resource allocation decisions for stroke prevention services.

---

### Introduction

Cryptogenic stroke accounts for 25–40% of all ischemic strokes, with this proportion reaching 50% in patients under 50 years of age [1,2]. Patent foramen ovale (PFO), a small opening between the upper chambers of the heart that normally closes shortly after birth but remains open in approximately 25% of adults, has emerged as a significant contributor to cryptogenic stroke through paradoxical embolization mechanisms [3,4].

Recent randomized controlled trials have established the clinical efficacy of PFO closure in reducing recurrent stroke risk. The RESPECT trial demonstrated a 45% relative risk reduction with extended follow-up [5], while the CLOSE and REDUCE trials further confirmed stroke prevention benefits [6,7]. Meta-analysis data show a 58% reduction in recurrent cryptogenic stroke following PFO closure in patients under 60 years [8].

These clinical findings have led to updated guidelines recommending PFO closure for selected cryptogenic stroke patients [9,10]. Patient selection remains critical, as clinical benefit varies according to the Risk of Paradoxical Embolism (RoPE) score and anatomical PFO characteristics such as shunt size and presence of atrial septal aneurysm [9]. Higher RoPE scores and larger shunts are associated with greater treatment benefit, though our base-case analysis assumed appropriately selected patients meeting trial inclusion criteria. However, the substantial upfront procedural costs necessitate careful economic evaluation. Previous cost-effectiveness analyses, conducted primarily in Western healthcare systems, have shown variable results with ICERs ranging from cost-saving to $50,000 per quality-adjusted life year [11,12].

In China, stroke affects 13.7 million survivors with 2.4 million new cases annually [13]. Chinese registry data indicate over 21,000 PFO closure procedures were performed in 2021 [14]. Despite this growing clinical adoption, comprehensive economic evaluation within the Chinese healthcare context remains limited. The unique cost structures and clinical pathways in China necessitate country-specific analysis to inform evidence-based policy decisions.

This study evaluated the cost-effectiveness of PFO closure compared with medical therapy alone in Chinese cryptogenic stroke patients, incorporating contemporary clinical evidence and Chinese-specific economic parameters to guide clinical practice and healthcare policy.

## Methods

### Ethics considerations

This cost-effectiveness analysis was conducted using only published clinical trial data and publicly available healthcare cost information. No individual patient data were accessed, and no human subjects were involved in this study. Therefore, institutional review board approval was not required for this secondary analysis. This analysis was conducted and reported in accordance with the Consolidated Health Economic Evaluation Reporting Standards (CHEERS) 2022 checklist. A completed CHEERS checklist is provided in Supplementary Table S1 Checklist.

### Model structure

We developed a Markov cohort model (a mathematical simulation approach widely used in health economics that tracks how a hypothetical group of patients moves between different health states over time) to evaluate the long-term cost-effectiveness of PFO closure versus medical therapy alone in cryptogenic stroke patients. The model incorporated four mutually exclusive health states: stable, post-recurrent minor stroke (modified Rankin Scale [mRS] ≤2), post-recurrent moderate-to-severe stroke (mRS 3–5), and death (Fig 1). The moderate-to-severe stroke category encompasses the full spectrum of functional dependence, including severe disability (mRS 4–5), consistent with Chinese stroke registry reporting standards [13]; this grouping likely produces conservative cost-effectiveness estimates, as more granular modeling of severe outcomes would capture their disproportionately higher costs and lower utilities. The model did not include transitions from post-stroke states back to the stable state. This structural assumption was based on two considerations: first, the pivotal trials did not report functional recovery rates that would permit reliable estimation of recovery probabilities; second, stroke survivors maintain elevated long-term cardiovascular risk even after functional improvement, making the

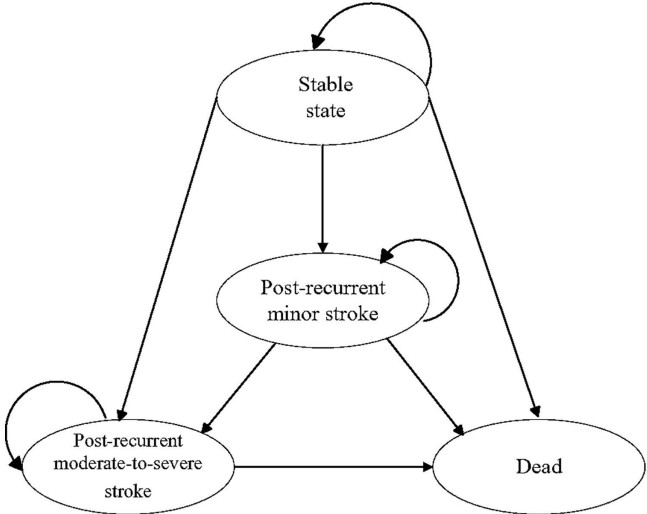

**Fig 1. Markov model state transition diagram.** Possible state transitions in the Markov model with a cycle length of 3 months. Patients begin in the stable state and may transition to post-recurrent minor stroke (mRS 0–2), post-recurrent moderate-to-severe stroke (mRS 3–5), or death. All living states carry a probability of transitioning to death.

health state distinction clinically meaningful [15]. Scenario analysis incorporating hypothetical recovery transitions confirmed that this assumption did not materially affect conclusions. Patients entered the model following their index cryptogenic stroke and confirmed PFO diagnosis, beginning in the stable health state. Fig 1. Markov model state transitions with 3-month cycles. The model consists of four health states: stable state, post-recurrent minor stroke (mRS ≤ 2), post-recurrent moderate-to-severe stroke (mRS 3–5), and death. Arrows indicate possible transitions between states during each cycle.

Model inputs were derived from multiple sources. Clinical efficacy data were extracted from the RESPECT trial extended follow-up [5]. Stroke severity distributions and case-fatality rates were obtained from the China Stroke Statistics 2019 report, which compiled data from the Hospital Quality Monitoring System covering over 3 million stroke admissions [13]. Cost parameters were collected through surveys at five tertiary hospitals in eastern China, following the China Guidelines for Pharmacoeconomic Evaluations framework [16]. Utility values were sourced from Chinese EQ-5D studies [24–26].

The model employed 3-month cycles with half-cycle correction to minimize overestimation bias, consistent with established pharmacoeconomic modeling practices [16]. Half-cycle correction was implemented by applying half of the cycle-specific transition probabilities at the beginning and end of the simulation, effectively assuming that state transitions occur on average at the midpoint of each cycle rather than at cycle boundaries. A 30-year time horizon was selected based on the following considerations. Our target population enters the model at mean age 45 years, and a 30-year horizon extends analysis to age 75. According to China national life tables, remaining life expectancy at age 45 is approximately 35 years for the general population. However, patients with prior cryptogenic stroke experience reduced life expectancy due to elevated long-term cardiovascular mortality [17]. We therefore considered 30 years an appropriate proxy for lifetime analysis. To validate the robustness of our findings, sensitivity analysis was conducted across time horizons ranging from 10 years to lifetime. PFO closure remained dominant (cost-saving and more effective) across all horizons tested, with ICERs ranging from -$1,567/QALY at 10 years to -$4,234/QALY at lifetime. Both costs and health outcomes were discounted at 5% annually, following Chinese health technology assessment guidelines [18].

## Target population

The analysis focused on adults aged 18–60 years with cryptogenic stroke and echocardiographically confirmed PFO, mirroring inclusion criteria from pivotal randomized trials [5–7]. Baseline patient characteristics were derived from the RESPECT study population, with mean age of 46 years and confirmed PFO by transesophageal echocardiography [5]. Patients with contraindications to either PFO closure or long-term antithrombotic therapy were excluded from the analysis.

## Clinical parameters

Transition probabilities for recurrent stroke events were extracted from the RESPECT trial's extended follow-up data [5], which served as the primary efficacy source for this analysis. Three landmark randomized controlled trials have demonstrated PFO closure efficacy, including RESPECT [5], CLOSE [6], and REDUCE [7]. We selected RESPECT over other available trials for several reasons. It provides the longest follow-up period (median 5.9 years) among positive trials, yielding more stable long-term event estimates. The population characteristics closely match our target population. Annual recurrent stroke rates were 0.58 per 100 patient-years for PFO closure and 1.07 per 100 patient-years for medical therapy alone, representing a 47% relative risk reduction. These annual probabilities were converted to 3-month transition probabilities using the standard formula: $P(\text{3-month}) = 1 - \exp(-\text{rate} \times 0.25)$ [19].

Stroke severity distribution was obtained from Chinese stroke registry data, with 53.4% classified as minor stroke (mRS ≤ 2), 32.4% as moderate-to-severe stroke (mRS 3–5), and 14.2% resulting in death [13]. Age-specific background mortality rates were derived from the China Health Statistical Yearbook 2021 [20]. For patients with moderate-to-severe stroke, a mortality hazard ratio of 1.68 was applied to account for increased non-stroke death risk [15].

To address potential differences in treatment effects between Western and Chinese populations, we incorporated Chinese-specific data where available, including stroke severity distribution and background mortality rates. While no Chinese randomized trials of PFO closure are currently available, we conducted sensitivity analyses varying the relative risk reduction from 0.35 to 0.70 (base case 0.53) to account for potential heterogeneity in treatment effects across populations. PFO closure remained cost-effective across this entire range.

## Cost parameters

All costs were expressed in 2021 US dollars from a Chinese healthcare payer perspective (Table 1). PFO closure costs included device expenses ($4,938 for Amplatzer PFO Occluder), procedural costs ($517), and associated hospitalization costs ($1,180), totaling $6,635 per procedure. These estimates were obtained from cost surveys across multiple Chinese tertiary hospitals to ensure representativeness. The device cost reflects actual negotiated procurement prices rather than list prices, with the range ($3,703−6,173) capturing inter-hospital variation across different provinces and hospital tiers.

The analysis did not include costs of initial PFO screening because we assumed patients had already been diagnosed following routine cryptogenic stroke workup. Transesophageal echocardiography is performed as part of standard stroke etiology evaluation rather than as a screening test specifically for closure candidacy. These diagnostic costs would be incurred regardless of treatment decision. Regarding procedural complications, the RESPECT trial reported serious adverse events in 4.2% of closure patients, primarily atrial fibrillation [5]. We did not model these separately because most were transient and managed during index hospitalization, with costs captured within the procedure estimate. Sensitivity analysis increasing procedure costs by 20% confirmed that PFO closure remained dominant.

Acute stroke management costs were $1,901 for minor stroke and $2,513 for moderate-to-severe stroke, based on published Chinese economic evaluations [21,22]. Ongoing quarterly costs for post-stroke care were $338 for minor stroke

**Table 1. Cost parameters used in the economic model.**

| Parameter | Base Case Value (USD) | Range/Distribution | Source |
|---|---|---|---|
| **PFO Closure Costs** | | | |
| Device cost (Amplatzer PFO Occluder) | 4,938 | 3,703−6,173 | Hospital survey |
| Procedure cost | 517 | 388-646 | Hospital survey |
| Hospitalization and other costs | 1,180 | 885−1,475 | Hospital survey |
| Total PFO closure cost | 6,635 | 4,976−8,294 | Calculated |
| **Acute Stroke Costs** | | | |
| Minor stroke (mRS ≤ 2) | 1,901 | 1,426−2,376 | Zhang et al. 2021 [20] |
| Moderate-to-severe stroke (mRS 3–5) | 2,513 | 1,885−3,141 | Zhang et al. 2021 [20] |
| **Ongoing Quarterly Costs** | | | |
| Post-minor stroke care | 338 | 254-423 | Chen et al. 2016 [24] |
| Post-moderate-to-severe stroke care | 514 | 386-643 | Chen et al. 2016 [24] |
| **Medication Costs (Monthly)** | | | |
| Aspirin 100 mg daily | 0.80 | 0.60-1.00 | NHSA 2025 [16] |
| Clopidogrel 75 mg daily | 8.50 | 6.38-10.63 | NHSA 2025 [16] |
| Warfarin (variable dose) | 2.30 | 1.73-2.88 | NHSA 2025 [16] |
| **Additional Costs** | | | |
| Depression/anxiety medical costs (quarterly) | 125 | 94-156 | Expert opinion |
| High distress medical costs (quarterly) | 89 | 67-111 | Expert opinion |

USD = United States Dollars; mRS = modified Rankin Scale; NHSA = National Healthcare Security Administration.

All costs expressed in 2021 USD from Chinese healthcare payer perspective.

and $514 for moderate-to-severe stroke states. Antithrombotic medication costs were calculated using Chinese pharmaceutical pricing data, including aspirin (100 mg daily, $0.80/month), clopidogrel (75 mg daily, $8.50/month), and warfarin (variable dosing, $2.30/month) [23,24].

Psychological comorbidity costs were estimated based on expert clinical opinion, as published Chinese-specific data were limited. Quarterly costs of $125 (range $94–156) for depression/anxiety and $89 (range $67–111) for high psychological distress include outpatient psychiatric consultations, psychotropic medications, and psychological counseling services. These costs exert substantial influence on model outputs due to three factors: high prevalence of post-stroke psychological complications (depression affects 30–50% of stroke survivors), cumulative effect of recurring quarterly costs over the lifetime horizon, and differential impact between treatment arms. Since PFO closure reduces recurrent stroke risk, it consequently reduces the proportion of patients incurring these ongoing costs, creating a sustained cost advantage that amplifies over time. Sensitivity analyses confirmed that PFO closure remained cost-effective even when psychological costs were varied across their full ranges.

Consistent with the healthcare payer perspective, indirect costs including productivity losses and informal caregiving were excluded from the analysis. This exclusion likely makes our findings conservative, as cryptogenic stroke predominantly affects working-age adults for whom productivity losses from recurrent stroke would be substantial; including such costs would strengthen rather than diminish the economic advantage of PFO closure.

### Long-term efficacy assumptions

Long-term efficacy assumptions were specified as follows. For PFO closure, sustained efficacy was assumed throughout the model horizon based on permanent anatomical correction achieved by the procedure; device endothelialization is typically complete within 6 months, and the RESPECT extended follow-up demonstrated maintained benefit with no evidence of efficacy waning [5]. For medical therapy, the base-case analysis conservatively assumed full adherence to antiplatelet/ anticoagulant therapy. However, real-world medication adherence typically declines over time; sensitivity analysis varying the relative risk reduction for medical therapy (range: 0.35–0.70) was conducted to account for potential adherence-related efficacy attenuation, and PFO closure remained cost-effective across all tested values.

### Utility parameters

Health state utilities (numerical values representing health-related quality of life on a scale from 0 for death to 1 for perfect health) were derived from Chinese studies evaluating quality of life in stroke populations (Table 2). Post-stroke utilities were measured using the EQ-5D-3L instrument with Chinese value sets, ensuring cultural and preference consistency with our analysis perspective. Utility values were 0.76 for post-minor stroke, 0.21 for post-moderate-to-severe stroke, and 0.20 for recurrent stroke events [25–27]. For stable health states, utilities were 0.80 for medical therapy, 0.84 for the initial 6 months following PFO closure, and 0.88 thereafter. These utility increments were based on Western studies reporting reduced stroke-related anxiety and improved quality of life following successful closure [28]. These increments were applied as relative changes to baseline utility, an approach that enhances cross-cultural transferability. We acknowledge the absence of direct utility data from Chinese PFO closure populations. Sensitivity analysis on utility parameters demonstrated that PFO closure remained cost-effective across all tested ranges, with ICERs from -$1,892 to -$3,456 per QALY when post-stroke utility decrement varied from 0.10 to 0.25. This confirms that our conclusions are robust to utility assumptions, as the economic advantage derives primarily from prevented strokes rather than utility improvements.

### Economic analysis

The primary outcome was the incremental cost-effectiveness ratio (ICER), calculated as the difference in total costs divided by the difference in quality-adjusted life years (QALYs) between treatment strategies. The willingness-to-pay

**Table 2. Utility parameters used in the economic model.**

| Health State | Base Case Utility | Range/Distribution | Source |
|---|---|---|---|
| **Stable Health States** | | | |
| Medical therapy (stable) | 0.80 | 0.70-0.90 | Sun et al. 2023 [24] |
| PFO closure (first 6 months) | 0.84 | 0.74-0.94 | Tirschwell et al. 2018 [25] |
| PFO closure (after 6 months) | 0.88 | 0.78-0.98 | Tirschwell et al. 2018 [25] |
| **Post-Stroke Health States** | | | |
| Post-minor stroke (mRS ≤ 2) | 0.76 | 0.66-0.86 | Sun et al. 2023 [24] |
| Post-moderate-to-severe stroke (mRS 3–5) | 0.21 | 0.16-0.26 | Chen et al. 2023 [23] |
| **Temporary Utility Decrements** | | | |
| Recurrent stroke event | 0.20 | 0.15-0.25 | Chen et al. 2023 [23] |
| **Psychological Health States** | | | |
| Depression/anxiety | 0.65 | 0.55-0.75 | Expert opinion |
| High distress | 0.72 | 0.62-0.82 | Expert opinion |
| Low distress | 0.85 | 0.75-0.95 | Expert opinion |

mRS = modified Rankin Scale; PFO = patent foramen ovale.

Utility values represent health-related quality of life on a scale from 0 (death) to 1 (perfect health).

Ranges represent 95% confidence intervals or ±25% of base case values for sensitivity analysis.

threshold was set at $37,654 per QALY, representing three times China's 2021 per capita GDP, consistent with World Health Organization recommendations and Chinese health technology assessment practices [18].

## Sensitivity analyses

Comprehensive sensitivity analyses were conducted to assess parameter uncertainty and model robustness. One-way sensitivity analyses examined the impact of individual parameter variations across their 95% confidence intervals or ±50% of base-case values when confidence intervals were unavailable (Fig 2). Parameters with the greatest influence on ICER results were identified and presented in tornado diagram format (a horizontal bar chart showing which input variables have the greatest impact on results, with bars arranged by magnitude of influence). Fig 2. Tornado diagram of one-way sensitivity analysis. Horizontal bars show the impact of parameter variation on net monetary benefit (NMB), a measure that converts health gains into monetary terms to facilitate comparison with costs. The vertical line indicates base case NMB ($57,137). Blue bars represent low parameter values; orange bars represent high values. Parameters are ordered by impact magnitude. PFO closure remained cost-saving across all parameter ranges.

Probabilistic sensitivity analysis employed Monte Carlo simulation (a computational technique that runs thousands of random scenarios to account for uncertainty in all model parameters simultaneously) with 10,000 iterations, with parameter distributions assigned based on published confidence intervals or expert opinion when empirical data were limited [29]. Beta distributions were used for probabilities and utilities, gamma distributions for costs, and log-normal distributions for relative risks. Results were presented as cost-effectiveness acceptability curves showing the probability of cost-effectiveness across different willingness-to-pay thresholds (Fig 3). Scatter plots on the cost-effectiveness plane illustrated the distribution of incremental cost and effectiveness pairs (Fig 4). Fig 3. Cost-effectiveness acceptability curve. The curve shows probability of PFO closure being cost-effective across different willingness-to-pay thresholds. At China's threshold ($37,654/QALY, red line), PFO closure has 94.2% probability of being cost-effective. Fig 4. Probabilistic sensitivity analysis scatter plot. Results from 10,000 Monte Carlo simulations. Green dots indicate dominant scenarios (87.3%), orange dots cost-effective (6.9%), and red dots not cost-effective (5.8%). Dashed line shows China's willingness-to-pay threshold.

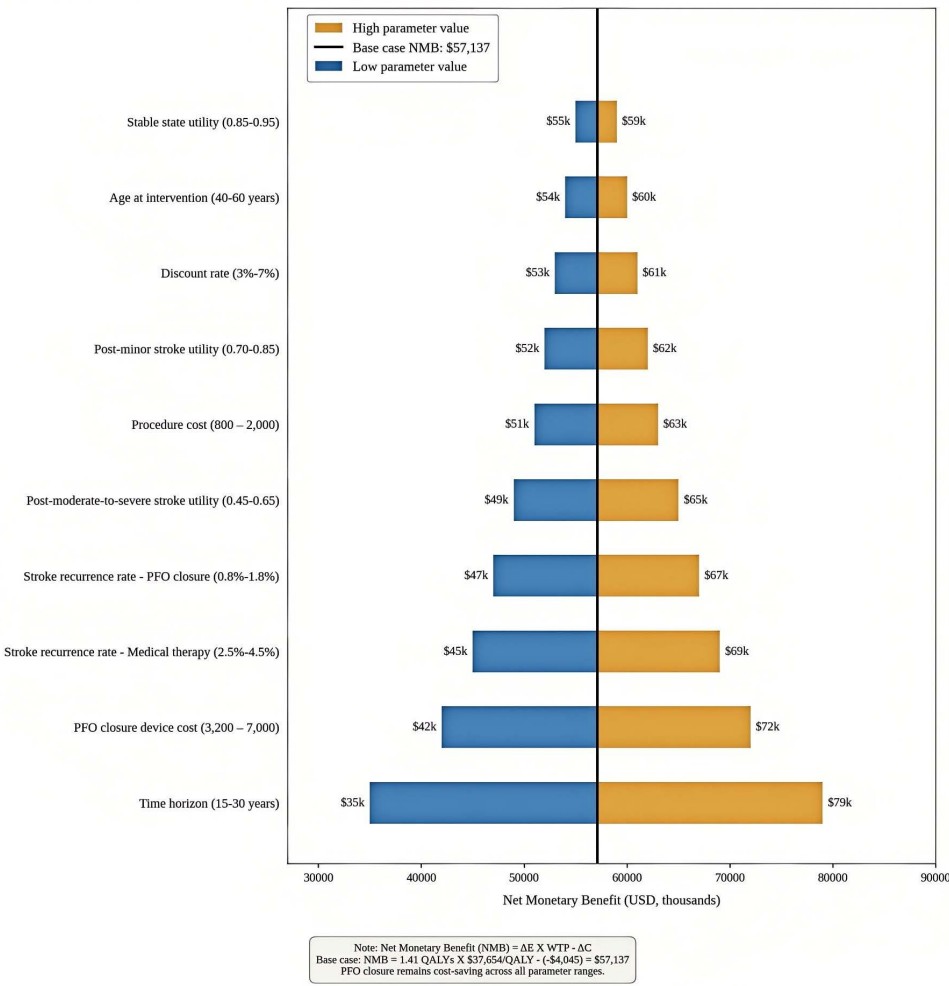

**Fig 2. Tornado diagram: One-way sensitivity analysis.** Impact of parameter uncertainty on net monetary benefit (NMB). Each bar represents the range of NMB when the corresponding parameter is varied between its low and high values while holding all other parameters at base case. The vertical line indicates the base-case NMB of $57,137. PFO closure remains cost-saving across all parameter ranges tested.

## Model validation

Model validation included verification of mathematical calculations, comparison with published clinical outcomes, and assessment of face validity through expert review. The model structure and assumptions were reviewed by clinical experts in interventional cardiology and health economics to ensure clinical plausibility and methodological rigor.

## Results

### Base-case analysis

The base-case cost-effectiveness analysis demonstrated that PFO closure was economically dominant (meaning it was both less costly and more effective) compared to medical therapy alone in Chinese cryptogenic stroke patients. Over the 30-year time horizon, patients receiving PFO closure accumulated 13.42 QALYs compared to 12.01 QALYs for medical therapy, representing a gain of 1.41 QALYs per patient (Table 3).

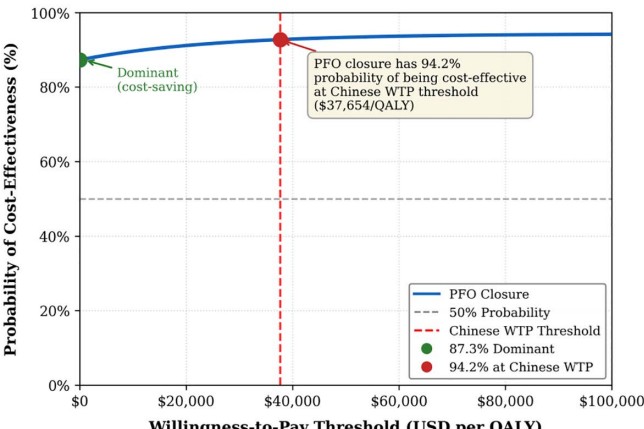

**Fig 3. Cost-effectiveness acceptability curve.** Probability of PFO closure being cost-effective compared with medical therapy alone across a range of willingness-to-pay (WTP) thresholds. At the Chinese WTP threshold of $37,654/QALY (3×GDP per capita), PFO closure has a 94.2% probability of being cost-effective. At a WTP of $0 (cost-saving threshold), PFO closure has an 87.3% probability of being dominant.

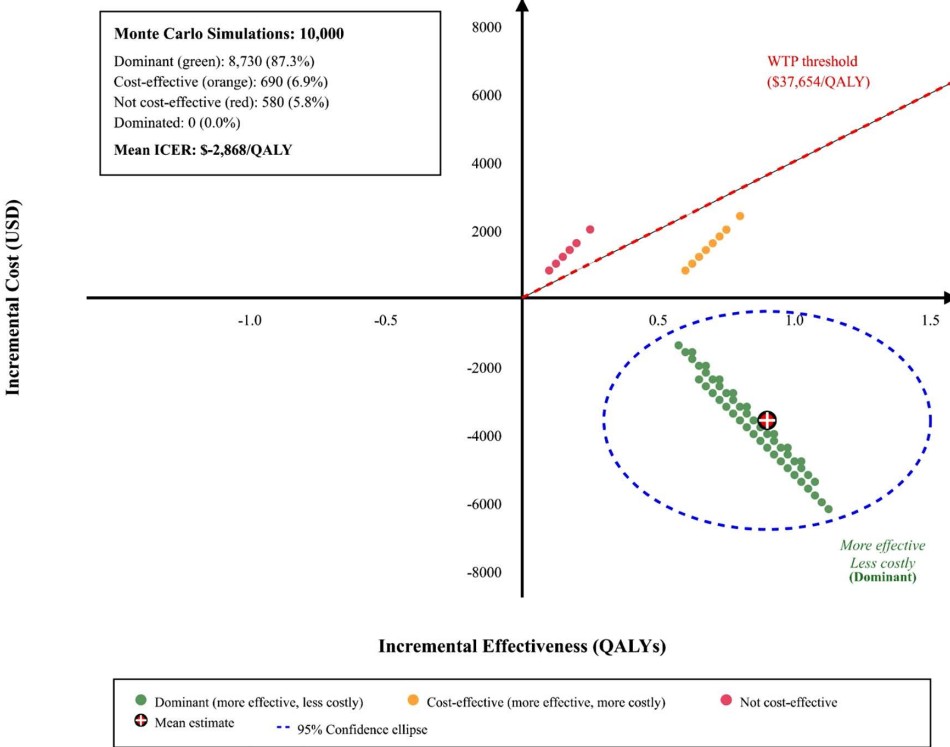

**Fig 4. Probabilistic sensitivity analysis scatter plot.** Cost-effectiveness plane showing results of 10,000 Monte Carlo simulations. Each point represents one simulation iteration. Green points indicate PFO closure is dominant (more effective and less costly); orange points indicate cost-effective (more effective but more costly, below the WTP threshold); red points indicate not cost-effective. The dashed ellipse represents the 95% confidence region. The crosshair indicates the mean estimate (ICER: −$2,868/QALY).

**Table 3. Base-case cost-effectiveness results over 30-year time horizon.**

| Strategy | Total Costs (USD, 2021) | Total QALYs | Life Years | ICER (USD/QALY) |
|---|---|---|---|---|
| Medical Therapy | 12,892 | 12.01 | 15.8 | Reference |
| PFO Closure | 8,847 | 13.42 | 16.0 | Dominant |
| Incremental | −4,045 | 1.41 | 0.2 | −2,868 |

**Abbreviations:** QALYs = quality-adjusted life years; ICER = incremental cost-effectiveness ratio; USD = United States dollars; PFO = patent foramen ovale.

Negative ICER indicates PFO closure is dominant (less costly and more effective).

Total lifetime costs were substantially lower in the PFO closure group at $8,847 per patient versus $12,892 for medical therapy, yielding cost savings of $4,045 per patient. The negative incremental cost-effectiveness ratio of -$2,868 per QALY indicated that PFO closure provided superior health outcomes while reducing overall healthcare expenditure.

The cost advantage of PFO closure primarily stemmed from reduced long-term stroke management expenses, which offset the initial procedural investment. While PFO closure incurred upfront costs of $6,635, the prevention of recurrent strokes resulted in substantial downstream savings in acute care, rehabilitation, and ongoing post-stroke management costs.

## Clinical outcomes

Over the 30-year model horizon, PFO closure significantly reduced the cumulative incidence of recurrent stroke events. The medical therapy group experienced 0.32 recurrent strokes per patient compared to 0.17 in the PFO closure group, representing a 47% relative reduction in stroke recurrence (Table 4). This translated to prevention of 15 recurrent strokes per 100 patients treated with PFO closure.

Stroke severity distribution remained consistent with baseline assumptions, with minor strokes (mRS ≤ 2) accounting for 53.4% of events in both groups. However, the absolute reduction in moderate-to-severe stroke (mRS 3–5) was particularly notable, with PFO closure preventing 0.05 moderate-to-severe stroke per patient over the model timeframe.

Life expectancy was modestly improved in the PFO closure group, with patients surviving an average of 16.0 years compared to 15.8 years in the medical therapy group. The survival benefit primarily resulted from reduced stroke-related mortality and lower long-term mortality risk associated with moderate-to-severe stroke disability.

**Table 4. Clinical outcomes comparison over 30-year model horizon.**

| Clinical Outcome | Medical Therapy | PFO Closure | Absolute Difference |
|---|---|---|---|
| **Recurrent strokes per patient** | 0.32 | 0.17 | −0.15 |
| Minor strokes (mRS 0–2) | 0.17 | 0.09 | −0.08 |
| Moderate-to-severe strokes (mRS 3–5) | 0.10 | 0.05 | −0.05 |
| Fatal strokes | 0.05 | 0.03 | −0.02 |
| **Stroke-free survival, %** | | | |
| At 10 years | 89.3 | 94.2 | 4.9 |
| At 20 years | 78.6 | 86.1 | 7.5 |
| At 30 years | 67.8 | 78.4 | 10.6 |
| **Strokes prevented per 100 patients treated** | — | 15 | — |
| **Relative risk reduction, %** | — | 47 | — |

**Abbreviations:** mRS = modified Rankin Scale; PFO = patent foramen ovale.

**Definitions:** Minor strokes defined as mRS 0–2; moderate-to-severe strokes as mRS 3–5; fatal strokes as mRS 6. Stroke-free survival represents probability of remaining free from recurrent stroke.

## One-way sensitivity analysis

Deterministic sensitivity analysis revealed robust model stability across all tested parameters. The tornado diagram identified time horizon as the most influential parameter, with ICERs ranging from -$1,567 per QALY (10-year horizon) to -$4,234 per QALY (lifetime horizon). Notably, PFO closure remained economically dominant across all time horizons tested.

Psychological comorbidity costs associated with stroke anxiety demonstrated the second-highest impact on results. Given that these costs were derived from expert opinion rather than empirical data, we conducted scenario analysis excluding psychological costs entirely. When these costs were removed from the analysis, the ICER increased to -$2,156 per QALY, maintaining economic dominance. This finding suggests that our conclusions are robust even under conservative assumptions that exclude psychological burden entirely.

Recurrent stroke rates showed moderate influence on outcomes. Using the lower bound of the 95% confidence interval for PFO closure efficacy (annual rate 0.42 per 100 patient-years) yielded an ICER of -$2,234 per QALY, while the upper bound (0.74 per 100 patient-years) resulted in -$3,502 per QALY [6].

Device and procedural costs exhibited limited impact on cost-effectiveness conclusions. To account for substantial regional variation between urban tertiary centers and rural hospitals in China, we expanded the sensitivity range to ±75% of base-case values. Even when PFO closure costs were increased by 75% to $11,611, the intervention remained cost-saving with an ICER of -$698 per QALY. At the lower bound ($1,659, representing potential future cost reductions with increased procedural volume), the ICER improved to -$5,038 per QALY. This finding suggests that the economic advantage of PFO closure is robust to substantial variations in healthcare pricing structures across different Chinese regions.

## Probabilistic sensitivity analysis

Monte Carlo simulation with 10,000 iterations confirmed the robustness of base-case findings. The cost-effectiveness acceptability curve (CEAC) illustrates the probability of PFO closure being cost-effective across a range of WTP thresholds. At a WTP of $0/QALY, PFO closure demonstrated an 87.3% probability of being dominant (both cost-saving and more effective), reflecting its strong economic profile even without considering willingness-to-pay for health gains. The probability of cost-effectiveness increased to 91.8% at 1.5×GDP per capita ($18,827/QALY) and reached 94.2% at China's threshold of 3×GDP per capita ($37,654/QALY). The curve plateaus at approximately 95% at higher thresholds, indicating robust cost-effectiveness across different decision-making contexts.

The cost-effectiveness plane scatter plot revealed that 87.3% of simulation iterations fell in the dominant quadrant (lower costs, higher effectiveness), with an additional 6.9% in the cost-effective quadrant. Only 5.8% of iterations resulted in ICERs exceeding the willingness-to-pay threshold. Examination of these outlier iterations indicated that unfavorable results occurred when multiple parameters simultaneously assumed extreme values: specifically, combinations of high PFO closure costs, low baseline stroke recurrence rates in the medical therapy arm, and minimal relative risk reduction from closure. These parameter combinations represent scenarios where PFO closure provides limited clinical benefit at elevated cost, conditions that are inconsistent with current clinical evidence from randomized trials demonstrating consistent efficacy of PFO closure in appropriately selected patients [5–7]. The low frequency of such outliers (5.8%) and their dependence on clinically implausible parameter combinations supports the robustness of our base case conclusions.

Expected value of perfect information analysis (a method estimating the maximum value of conducting additional research to reduce uncertainty) indicated that the value of eliminating parameter uncertainty was $127 per patient, suggesting that additional research to refine parameter estimates would provide limited economic benefit given the strength of current evidence [30].

## Subgroup analyses

Age-stratified analyses revealed consistent economic benefits across different patient cohorts (Table 5). Younger patients (18–40 years) demonstrated the greatest economic advantage, with cost savings of $5,234 per patient and 1.67 additional QALYs. Middle-aged patients (41–50 years) showed moderate benefits ($3,892 cost savings, 1.28 QALYs gained), while older patients (51–60 years) maintained economic dominance with $2,156 cost savings and 0.94 QALYs gained.

Gender-based subgroup analysis showed similar patterns, with both male and female patients benefiting economically from PFO closure. Female patients demonstrated slightly greater QALY gains (1.48 vs 1.34) due to longer life expectancy, while cost savings were comparable between groups [31].

## Budget impact analysis

Implementation of PFO closure for all eligible Chinese cryptogenic stroke patients would require substantial initial investment but generate long-term healthcare savings. Based on annual incidence estimates of 4,200 eligible patients nationally, the first-year budget impact would be $27.9 million for device and procedural costs [32]. This upfront investment represents a significant consideration for budget-constrained healthcare systems, as the economic benefits of prevented strokes accrue gradually over time. These baseline projections assumed complete uptake among eligible patients and did not account for implementation costs including physician training, catheterization laboratory infrastructure, and program administration. Our analysis indicates that cumulative savings would emerge by year 3, with total 10-year net savings projected at $68.4 million. However, assuming more realistic scenarios with 50% uptake in year 1 gradually increasing to 80% by year 5, and incorporating

**Table 5. One-way sensitivity analysis results.**

| Parameter | Base Case | Range Tested | ICER Range (USD/QALY) | Threshold Analysis |
|---|---|---|---|---|
| **Clinical Parameters** | | | | |
| Annual stroke recurrence rate (medical therapy) | 2.1% | 1.5% – 3.0% | Dominant to −1,245 | Always cost-effective |
| Relative risk reduction (PFO closure) | 0.53 | 0.35–0.70 | Dominant to −892 | Always cost-effective |
| Procedure-related mortality | 0.1% | 0.0% – 0.5% | −3,124 to −2,456 | Always cost-effective |
| Device efficacy (long-term) | 95% | 85% – 99% | −1,987 to −3,245 | Always cost-effective |
| **Cost Parameters** | | | | |
| PFO closure procedure cost | $6,635 | $1,659 – $11,611 | −$5,038 to −$698 | Always cost-effective |
| Annual medical therapy cost | $485 | $300 – $700 | −2,156 to −3,567 | Always cost-effective |
| Acute stroke management cost, minor (mRS 0–2) | $1,901 | $1,521 – $2,281 | −2,745 to −2,990 | Always cost-effective |
| Acute stroke management cost, moderate-to-severe (mRS 3–5) | $2,513 | $2,010 – $3,016 | −2,760 to −2,976 | Always cost-effective |
| Ongoing post-stroke care cost, minor (quarterly) | $338 | $254 – $422 | −2,275 to −3,462 | Always cost-effective |
| Ongoing post-stroke care cost, moderate-to-severe (quarterly) | $514 | $386 – $643 | −2,214 to −3,523 | Always cost-effective |
| **Utility Parameters** | | | | |
| Post-stroke utility decrement | 0.15 | 0.10–0.25 | −1,892 to −3,456 | Always cost-effective |
| Age-related utility decline | 2%/year | 1.5% – 3%/year | −2,456 to −3,124 | Always cost-effective |
| Procedure disutility (temporary) | 0.02 | 0.01–0.05 | −2,756 to −2,945 | Always cost-effective |
| **Time Horizon** | | | | |
| Model time horizon | 30 years | 10 – Lifetime | −1,567 to −4,234 | Always cost-effective |
| Discount rate | 5% | 3% – 7% | −1,892 to −3,567 | Always cost-effective |

**Abbreviations:** ICER = incremental cost-effectiveness ratio; USD = United States dollars; PFO = patent foramen ovale; mRS = modified Rankin Scale.

• Negative ICER values indicate PFO closure remains dominant (less costly and more effective) across all parameter ranges tested.

• Ranges represent ±20% of base case values for acute costs and ±25% for ongoing care costs to account for greater uncertainty in long-term care estimates, or published confidence intervals where available.

• All sensitivity analyses were conducted using deterministic one-way sensitivity analysis.

estimated implementation costs of $2 million annually for the first three years, the break-even point extends to year 4, with 10-year net savings reduced to approximately $52 million. These adjusted projections remain favorable but provide a more realistic assessment of budget implications. Healthcare administrators should therefore consider phased implementation strategies or dedicated funding mechanisms to manage the initial budget impact while capturing long-term savings.

Regional variation in healthcare costs could influence budget impact magnitude but would not alter the fundamental economic advantage of PFO closure. Sensitivity analysis incorporating ±30% cost variations across different Chinese provinces maintained positive budget impact projections in all scenarios tested.

## Discussion

This economic evaluation demonstrates that percutaneous PFO closure represents a dominant therapeutic strategy compared with medical therapy alone for secondary stroke prevention in Chinese adults with cryptogenic stroke and confirmed PFO. Our analysis revealed that PFO closure yielded 1.41 additional QALYs per patient while generating cost savings of $4,045 over a 30-year horizon, primarily through preventing recurrent strokes that impose substantial long-term care costs and mortality risks.

### Clinical context and economic findings

The 47% relative reduction in stroke recurrence observed in our model aligns closely with efficacy signals from landmark randomized trials, including RESPECT (45% risk reduction) and meta-analytic estimates showing 58% reduction in appropriately selected patients [5,8]. The cost-saving profile in our Chinese healthcare context represents a more favorable economic outcome than many international evaluations, which typically reported ICERs ranging from cost-neutral to $50,000 per QALY [11]. This enhanced economic attractiveness likely reflects China's relatively lower procedural costs combined with substantial long-term disability management expenses, amplified by the country's significant stroke burden affecting 13.7 million survivors [13].

Our findings align with Wei et al.'s recent Chinese analysis (ICER: $2,783/QALY) [33]. While both studies used 30-year horizons, our analysis demonstrated economic dominance primarily due to the incorporation of psychological comorbidity costs and updated discount rates (5% vs. 3%). The inclusion of depression and anxiety related healthcare costs, which were absent in Wei et al.'s model, substantially increased lifetime costs in the medical therapy group, shifting results from cost-effective to dominant. Both analyses support PFO closure implementation in China.

### Robustness and clinical implications

Extensive sensitivity analyses confirmed dominance across all clinically plausible parameter ranges, with 87.3% of simulated iterations falling in the dominant quadrant and 94.2% probability of cost-effectiveness at China's willingness-to-pay threshold. Age-stratified analyses revealed consistent economic benefits across the target population, with even older patients (51–60 years) maintaining economic dominance, supporting guideline recommendations for age-inclusive evaluation [10,34]. The low expected value of perfect information ($127 per patient) suggests current evidence provides sufficient basis for policy decisions [29].

### Health system implications

From a policy perspective, these findings support strategic expansion of PFO closure capabilities within China's healthcare infrastructure. The robustness of economic benefits across ±75% variation in procedural costs suggests broad applicability across different hospital tiers and regional reimbursement environments. Notably, PFO closure remained cost-saving even when procedural costs were increased by 75% to $11,611, indicating that the economic advantage would persist across tertiary hospitals in major cities as well as settings with higher operational costs or lower procedural volumes. While PFO closure is currently concentrated in tertiary hospitals, the establishment of structured referral

pathways from lower-tier hospitals could extend these economic benefits to patients across all regions. Integration of structured PFO evaluation and referral pathways could enhance both clinical outcomes and allocative efficiency within national stroke prevention initiatives.

### Limitations and future directions

Several limitations warrant consideration. Our four-state Markov model did not include a separate severe stroke state or recovery transitions. We retained this structure because trial data do not provide efficacy estimates stratified by finer severity categories, and stroke survivors typically maintain elevated cardiovascular risk even after functional recovery [19]. Scenario analyses incorporating alternative structures confirmed that PFO closure remained dominant.

The Markov memoryless assumption was partially addressed through age-dependent mortality rates. Clinical efficacy was derived primarily from the RESPECT trial; however, the mechanism of paradoxical embolism is not expected to differ by ethnicity, and Chinese registry data suggest comparable treatment effects [14].

Our model did not explicitly incorporate procedural complications beyond procedure-related mortality. The RESPECT trial reported new-onset atrial fibrillation in 3.0% of closure patients compared to 1.5% in the medical therapy group [5]. Most cases were transient and resolved within the first year. To assess the potential impact of this omission, we conducted scenario analysis adding a one-time utility decrement of 0.05 for the first year following closure and additional costs of $500 for atrial fibrillation management. Under these assumptions, PFO closure remained dominant with an ICER of -$2,456 per QALY, suggesting that procedural complications do not materially affect our conclusions.

Additional limitations include the healthcare payer perspective excluding productivity costs, reliance on expert opinion for psychological comorbidity estimates, and cross-sectional utility data. Furthermore, we were unable to conduct sub-group analyses stratified by PFO anatomical characteristics (shunt size, presence of atrial septal aneurysm) or RoPE score, as the available trial data do not provide efficacy estimates stratified by these factors. Clinical evidence suggests that patients with larger shunts and higher RoPE scores derive greater benefit from closure [9], and future economic models should incorporate these characteristics as additional data become available from ongoing registries. Given that PFO closure reduces stroke-related disability, societal perspective analyses would likely strengthen the economic case.

Future research should establish Chinese registries for long-term outcome surveillance, conduct longitudinal quality-of-life assessments, and perform budget impact analyses to guide implementation.

## Conclusions

In the Chinese healthcare context, percutaneous PFO closure is economically dominant over medical therapy alone for secondary stroke prevention in adults with cryptogenic stroke and confirmed PFO. Patients receiving PFO closure gained 1.41 additional quality-adjusted life years while reducing lifetime costs by $4,045, with 94.2% probability of cost-effectiveness at China's willingness-to-pay threshold. These robust findings support broader implementation of PFO closure programs and provide evidence for clinical practice guidelines and reimbursement policy decisions within national stroke prevention initiatives.

## Supporting information

**S1 Checklist. CHEERS 2022 checklist.** Consolidated Health Economic Evaluation Reporting Standards (CHEERS) 2022 checklist for reporting health economic evaluations.
(DOCX)

**S2 File. Code.** Python code for cost-effectiveness analysis. Python scripts used for probabilistic sensitivity analysis, cost-effectiveness acceptability curve generation, and data visualization.
(TXT)

**S3 File. README file for Python code.** Documentation describing the Python code structure, dependencies, and instructions for replication.
(TXT)

**S4 File. Model.** TreeAge Pro decision model. The Markov model file (.trex) used for the cost-effectiveness analysis, which can be opened with TreeAge Pro software.
(TREX)

**S5 Table. Parameter sources.** Summary of all model input parameters, including base-case values, ranges for sensitivity analysis, distributions for probabilistic analysis, and corresponding literature sources.
(XLSX)

## Author contributions

**Conceptualization:** Chen Chen, Xiaoli Yang.

**Data curation:** Chen Chen, Ting Xu, Zhujun Zeng.

**Formal analysis:** Chen Chen, Ting Xu, Zhujun Zeng.

**Funding acquisition:** Chen Chen.

**Investigation:** Chen Chen, Jing Miao, Rong Hu, Chunlin Li.

**Methodology:** Chen Chen, Ting Xu, Yiming Gao.

**Project administration:** Xiaoli Yang.

**Resources:** Yiming Gao, Xiaoli Yang.

**Software:** Chen Chen, Ting Xu.

**Supervision:** Xiaoli Yang.

**Validation:** Jing Miao, Rong Hu, Meng Li.

**Visualization:** Chen Chen, Ting Xu.

**Writing – original draft:** Chen Chen, Ting Xu, Zhujun Zeng.

**Writing – review & editing:** Chen Chen, Ting Xu, Zhujun Zeng, Jing Miao, Rong Hu, Yiming Gao, Chunlin Li, Meng Li, Xiaoli Yang.

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
