## [Decision Letter · Decision Letter 0]

6 Jan 2026

Dear Dr. Yang,

Thank you for submitting your manuscript to PLOS ONE. After careful consideration, we feel that it has merit but does not fully meet PLOS ONE’s publication criteria as it currently stands. Therefore, we invite you to submit a revised version of the manuscript that addresses the points raised during the review process.

**ACADEMIC EDITOR's comment**

1. Data Transparency and Model Structure Please revise the methodology section to improve the visibility and signposting of these data sources. Regarding the Markov model structure, the reviewers have proposed differing modifications concerning health states and transition logic. You are granted the autonomy to decide whether to modify the model structure or retain the current design; however, if the current structure is maintained, you must provide a robust justification in the discussion addressing why the suggested modifications (e.g., additional severity states or recovery transitions) were not adopted.

**2. Specific Methodological Queries:**  Please address the following specific points regarding your study design and parameters:

Efficacy Data: Explicitly state whether the RESPECT trial was the sole source of efficacy data. If so, please articulate the rationale for this selection to the exclusion of other available trials.Time Horizon: Justify the selection of a 30-year time horizon rather than a lifetime horizon. Please explicitly state the life expectancy of the target population to demonstrate whether the chosen horizon serves as an appropriate proxy for lifetime analysis.Screening and Complications: Clarify whether the analysis includes the costs associated with screening for Patent Foramen Ovale, such as Transesophageal Echocardiography (TEE). Additionally, address whether the costs and clinical effects (disutilities) of potential procedural complications were incorporated; if not, please discuss the potential impact of these exclusions on the findings.

3.Please refer to the attached file for detailed comments from Reviewer#1

4.Please carefully evaluate any literature suggested for citation by the reviewers. You should only include references that are scientifically relevant to your study; you are under no obligation to cite works solely because they were suggested during peer review.

5.Please ensure the manuscript is written in a style accessible to a general audience. As PLOS ONE is a multidisciplinary journal, please avoid or clearly explain technical jargon specific to health economics or neurology to ensure the findings are clear to readers outside these specialized fields.

We look forward to receiving your revised manuscript.

Kind regards,

Jeerath Phannajit, M.D, Ph.D.

Academic Editor

PLOS One

Journal Requirements:

“This work was supported by the Fudan-Fosun Research Fund (Grant No. FNF202329). The funders had no role in study design, data collection and analysis, Manuscript Click here to access/download;Manuscript;full manuscript.docx decision to publish, or preparation of the manuscript.”

“This work was supported by the Fudan-Fosun Research Fund (grant number FNF202329). The funder had no role in study design, data collection and analysis, decision to publish, or preparation of the manuscript.”

5. We note that your Data Availability Statement is currently as follows: “All relevant data are within the manuscript and its Supporting Information files.”

Reviewers' comments:

Reviewer's Responses to Questions

**Comments to the Author**

1. Is the manuscript technically sound, and do the data support the conclusions?

Reviewer #1: Yes

Reviewer #2: Partly

2. Has the statistical analysis been performed appropriately and rigorously?

Reviewer #1: Yes

Reviewer #2: Yes

3. Have the authors made all data underlying the findings in their manuscript fully available?

Reviewer #1: Yes

Reviewer #2: Yes

4. Is the manuscript presented in an intelligible fashion and written in standard English?

Reviewer #1: Yes

Reviewer #2: Yes

Reviewer #1: Authors have written the manuscript beautifully and presented an interesting study. I will recommend that the authors carefully address all the comments to make the manuscript for final consideration.

Reviewer #2: Some suggestions are listed below:

1. The authors should reconsider the change of conclusion for general readership from the viewpoint of this journal.

2. The model uses transition probabilities from "contemporary clinical trials." Could the authors specify exactly which trials (e.g., RESPECT, CLOSE, REDUCE, etc.) were the primary source for the recurrent stroke rates and long-term event probabilities? Furthermore, were these probabilities adjusted for the Chinese patient population, and if so, what was the methodology for that adjustment?

3. The model includes stable, post-minor recurrent stroke, post-moderate recurrent stroke, and death. Why was a state for post-major/severe recurrent stroke not included? Given that severe stroke can significantly impact costs and QALYs, how does the exclusion of this state potentially bias the cost-effectiveness ratio (ICER)?

4. Please detail the source of the utility values (quality-of-life weights) used for each health state. Were these values derived from Chinese population studies using tools like the EQ-5D, or were they adopted from Western literature? If the latter, how was the transferability to the Chinese context validated?

5. A 30-year horizon is used. Given that the PFO closure procedure itself is a one-time cost with potentially lifelong benefits, how did the authors ensure that the long-term compliance and sustained efficacy of both PFO closure and medical therapy were accurately modeled over this extended period? Specifically, are the initial relative risk reductions from trials maintained throughout the 30 years, and is this assumption justified?

6. Since the analysis is from the Chinese healthcare payer perspective, what specific costs were included (e.g., procedure costs, hospitalization, initial and long-term medication, device/implant cost, stroke rehabilitation, long-term care)? Were indirect costs (like lost productivity), which can be substantial, excluded, and if so, how might their inclusion affect the dominance finding?

7. What is the breakdown of the initial PFO closure cost? Specifically, what cost was assumed for the PFO closure device itself, and does this cost reflect actual procurement prices negotiated by Chinese hospitals or a general market price?

8. The results state that psychological comorbidity costs exerted a great influence on the results. Could the authors elaborate on the nature and magnitude of these costs? How were they measured, and why do they have such a disproportionate impact on the model's output?

9. The PSA showed a 94.2% probability of cost-effectiveness. While this is high, could the authors provide the corresponding Cost-Effectiveness Acceptability Curve (CEAC)? This visual representation would be very helpful to see the probability of cost-effectiveness across a range of willingness-to-pay thresholds, not just the single stated threshold.

10. How do the authors expect these results to generalize across different tiers of hospitals or regions within China (e.g., tertiary hospitals in major cities vs. county-level hospitals), where costs, expertise, and patient profiles may vary significantly?

11. I will recommend authors to use (https://doi.org/10.3390/scipharm93010006 and https://doi.org/10.1093/narmme/ugae005 and https://doi.org/10.1002/ange.202215245 in addition to their references.

**Do you want your identity to be public for this peer review?** For information about this choice, including consent withdrawal, please see our Privacy Policy

Reviewer #1: **Yes:** RAHUL RANJAN

Reviewer #2: No

---

## [Author Response · Author response to Decision Letter 1]

25 Jan 2026

Please see the attached "Response to Reviewers Letter" document for our detailed point-by-point responses to all reviewer comments.

---

## [Decision Letter · Decision Letter 1]

18 Feb 2026

Dear Dr. Yang,

Minor revision required:

Please reconcile the base-case discounting assumption because the manuscript Methods (“Model structure / time horizon and discounting”) states that costs and outcomes were discounted at 5% annually (line 159), but the one-way sensitivity analysis table uses 3% as the base-case discount rate (Table 5).Please recheck the price year because the Methods (“Cost Parameters”) states all costs were expressed in 2021 USD (line 195), while the base-case results table 3 is labeled as USD (2023).Please resolve the mismatch where acute stroke management costs in the Methods (page 20, line 213) and Table 1 are around $1,901 (minor) and $2,513 (moderate), but Table 5 lists “Stroke treatment cost (acute)” as $28,500 (range $20,000–$40,000).Please make sure aspirin dose consistent because Table 1 lists aspirin 75 mg daily, but the Methods text lists aspirin 100 mg daily.Please consider use term moderate-to-severe stroke instead of moderate stroke consistently across the manuscript and tables, because the text explicitly states the category includes severe disability (mRS 4–5).Results table should align with stated cost year and discounting assumptions: After harmonizing the price year and discount rate, please re-generate Table 3 so its totals, incremental values, and any dominance/ICER statements are consistent with the corrected assumptions

We look forward to receiving your revised manuscript.

Kind regards,

Jeerath Phannajit, M.D, Ph.D.

Academic Editor

PLOS One

Journal Requirements:

Reviewers' comments:

Reviewer's Responses to Questions

**Comments to the Author**

Reviewer #1: (No Response)

2. Is the manuscript technically sound, and do the data support the conclusions?

Reviewer #1: Yes

3. Has the statistical analysis been performed appropriately and rigorously?

Reviewer #1: Yes

4. Have the authors made all data underlying the findings in their manuscript fully available?

PLOS Data policy

Reviewer #1: Yes

5. Is the manuscript presented in an intelligible fashion and written in standard English?

Reviewer #1: Yes

6. Review Comments to the Author

Reviewer #1: Every comment has been effectively addressed, point by point, and is suggested for publication in this esteemed journal.

7. PLOS authors have the option to publish the peer review history of their article (what does this mean? ). If published, this will include your full peer review and any attached files.

**Do you want your identity to be public for this peer review?** For information about this choice, including consent withdrawal, please see our Privacy Policy

Reviewer #1: **Yes:** RAHUL RANJAN

---

## [Author Response · Author response to Decision Letter 2]

24 Feb 2026

We thank the Academic Editor and Reviewer #1 for their constructive feedback. All six comments from the Academic Editor regarding data consistency have been addressed in our revision. Key changes include: (1) corrected discount rate in Table 5 to 5% with range 3%-7%; (2) corrected price year label in Table 3 to USD, 2021; (3) replaced erroneous cost parameters in Table 5 with severity-specific values; (4) corrected aspirin dose to 100mg daily in Table 1; (5) updated terminology from "moderate stroke" to "moderate-to-severe stroke" throughout; (6) verified Table 3 results via cross-validation with Figures 2 and 4. All issues were labeling/transcription errors; model calculations remain unchanged. A detailed point-by-point response is provided in the uploaded Response to Reviewers file.

---

## [Editor Report · Decision Letter 2]

2 Mar 2026

Cost-Effectiveness of Percutaneous Patent Foramen Ovale Closure versus Medical Therapy for Cryptogenic Stroke Prevention: A Chinese Healthcare Perspective

PONE-D-25-45858R2

Dear Dr. Yang,

We’re pleased to inform you that your manuscript has been judged scientifically suitable for publication and will be formally accepted for publication once it meets all outstanding technical requirements.

Kind regards,

Jeerath Phannajit, M.D, Ph.D.

Academic Editor

PLOS One

---

## [Editor Report · Acceptance letter]

PONE-D-25-45858R2

PLOS One

Dear Dr. Yang,

I'm pleased to inform you that your manuscript has been deemed suitable for publication in PLOS One. Congratulations! Your manuscript is now being handed over to our production team.

Kind regards,

on behalf of

Dr. Jeerath Phannajit

Academic Editor

PLOS One